# Possible Association of Cholesterol as a Biomarker in Suicide Behavior

**DOI:** 10.3390/biomedicines9111559

**Published:** 2021-10-28

**Authors:** Thelma Beatriz González-Castro, Alma Delia Genis-Mendoza, Dulce Ivannia León-Escalante, Yazmín Hernández-Díaz, Isela Esther Juárez-Rojop, Carlos Alfonso Tovilla-Zárate, María Lilia López-Narváez, Alejandro Marín-Medina, Humberto Nicolini, Rosa Giannina Castillo-Avila, Miguel Ángel Ramos-Méndez

**Affiliations:** 1División Académica Multidisciplinaria de Jalpa de Méndez, Universidad Juárez Autónoma de Tabasco, Jalpa de Méndez 86205, Tabasco, Mexico; thelma.glez.castro@gmail.com (T.B.G.-C.); yazmin.hdez.diaz@gmail.com (Y.H.-D.); 2Departamento de Genética Psiquiátrica, Instituto Nacional de Medicina Genómica, Ciudad de México 14610, Mexico; adgenis@inmegen.gob.mx; 3División Académica Multidisciplinaria de Comalcalco, Universidad Juárez Autónoma de Tabasco, Comalcalco 86650, Tabasco, Mexico; ivanniaescalante10@gmail.com; 4División Académica de Ciencias de la Salud, Universidad Juárez Autónoma de Tabasco, Villahermosa 86100, Tabasco, Mexico; iselajuarezrojop@hotmail.com (I.E.J.-R.); gianninaavila2012@hotmail.com (R.G.C.-A.); angel_mar@live.com.mx (M.Á.R.-M.); 5Secretaría de Salud de Chiapas, Hospital Chiapas Nos Une “Dr. Gilberto Gómez Maza”, Tuxtla Gutiérrez 29045, Chiapas, Mexico; dralilialonar@yahoo.com.mx; 6Universidad de Guadalajara, CUCS, Guadalajara 44340, Jalisco, Mexico; stat5A@hotmail.com

**Keywords:** psychiatric illnesses, suicide, cholesterol, serum

## Abstract

Suicides and suicidal behavior are major causes of mortality and morbidity in public health and are a global problem. Various authors have proposed changes in lipid metabolism (total cholesterol decrease) as a possible biological marker for suicidal behavior. The objective of this study was to review the studies that have demonstrated a relationship between serum cholesterol levels and suicidal behavior and to describe the possible pathophysiological mechanisms that associate changes in cholesterol concentration and suicidal behavior. Relevant literature related to serum cholesterol levels and suicidal behavior was identified through various database searches. The data from the existing literature present the findings that relate low cholesterol levels and possible pathophysiological mechanisms (neuroinflammation, serotonergic neurotransmission), genes related to cholesterol synthesis, pharmacological treatments that alter lipid metabolism and the possible participation in suicidal behavior. Nevertheless, future research is required to describe how serum cholesterol affects cholesterol metabolism in the CNS to establish and understand the role of cholesterol in suicidal behavior.

## 1. Introduction

The term suicide encompasses ideation, intent, and actions. Suicidal ideation is defined as thinking about or considering suicide; ranging from passive death wishes to active thoughts of committing suicide. In this sense, suicide is listed by the World Health Organization as one of the leading causes of deaths worldwide. About 800,000 people take their own lives every year, which generates a global mortality rate of 16 per 100,000 inhabitants. Lifetime prevalence rates are approximately 9.2% for suicidal ideation and 2.7% for suicide attempts [1]. It is estimated that at least 20 people try to commit suicide per each individual who dies by suicide [2]. There are many variables that are related to suicidal behavior, some articles even propose an important role of climate change and environmental factors [3].

Cholesterol is a lipid that realizes many important functions in the cell, one of them is cell signaling, since cholesterol is essential for lipid rafts, structures necessary for multiple cellular functions [4]. Alterations in cholesterol metabolism mainly affect cardiovascular diseases, but also neurodegenerative diseases and even cancer, due to the role that cholesterol plays in metabolism [5]. The possible role of cholesterol in suicidal behavior is currently being investigated, however, the results are controversial, but many of these studies did not consider confounding variables that can affect cholesterol metabolism.

### Cholesterol in CNS

The composition of lipids in the CNS maintains the structure, functionality, and integrity of the cell membrane structures. Moreover, the main component in the central nervous system (CNS) is cholesterol (25% of the total body cholesterol) [6,7]. This is present in myelin sheaths of the white matter, the membranes of the glial cells, and neurons of the gray matter (cognitive functions). On the other hand, it is known that cholesterol metabolism in CNS is different than in the rest of the body, since the blood–brain barrier prevents the transit of cholesterol molecules; therefore, cholesterol in the brain is obtained from in situ biosynthesis, and transport of cholesterol between different cells. Cholesterol biosynthesis is through the Bloch (astrocytes) and the Kandutsch–Russell pathway (neurons).

The neurons have less ability to perform synthesis of cholesterol, they depend on astrocytes to perform this function. Once cholesterol has been synthesized in the endoplasmatic reticulum, it is transferred to the plasma membrane by binding to proteins (apolipoprotein E (ApoE) and ABC-like transporters), and lysosomes are responsible for converting it into free cholesterol, with the help of NPC1/NPC2. Additionally, the excess cholesterol is esterified intracellularly, stored as lipid droplets, eliminated in the form of oxysterol by the action of the enzyme 24-hydroxylase [8], this enzyme participates in the homeostasis of cholesterol in the CNS.

The average life of cerebral cholesterol is between 6 months and 5 years [7]. In this sense, changes in cholesterol concentrations in brain alter the CNS functions, a decrease of cholesterol at neuronal level produces alterations in neurotransmission and synaptic degeneration, as it has been described in several brain diseases including Huntington’s, Alzheimer’s, and Parkinson’s diseases [6,9]. Interestingly, increased levels of oxysterols have been observed in the prefrontal cortex in postmortem individuals who died by suicide. There is an association between low cholesterol levels and loss of memory, cognitive impairment [6], mood disorders, and suicide behavior [10,11]. The lower levels of cholesterol could result in synapsis alteration [12], favoring suicide behavior.

## 2. Association between Cholesterol and Suicidality

Several studies have investigated a possible link between low serum cholesterol and suicide. Zhang and et al. (2020) suggest that MDD patients who attempted suicide had significantly lower serum total cholesterol and LDL-C levels than those who did not attempt suicide. In this sense, Wu et al. (2016) [13] by means of a meta-analysis study in patients with depressive disorders (depression, schizoaffective depression, major depressive episodes, and MDD), report that patients with suicide attempt have lower serum levels of TC, TG HDL-C, LDL-C than non-suicidal patients. Moreover, the lower serum concentrations of TC present a relationship with the suicide attempt in middle-aged and older patients with respect to subjects under 40 years of age [14]. These findings together support the hypothesis that lower serum concentrations of CT and LDL-C are associated with suicide attempt in patients with MDD. In a study carried out in 52 patients hospitalized after a suicide attempt, several parameters were analyzed, a statistically significant association was found with the decrease in total cholesterol and HDL-C, a relationship was also observed between the decrease in total serum cholesterol and hostile personality traits [15]. Lester et al. (2002) [16] found that people with lower levels of serum cholesterol had a statistically significant rate of completion of suicide, compared to people who had serum cholesterol in normal ranges, and people who had attempted suicide in the past also had lower cholesterol levels, especially when they used violent suicide methods. Wu et al. (2016) [13] reported that lower serum cholesterol levels were associated with a 123% higher risk of suicide attempt and 85% higher risk of suicide completion. Recently, it was suggested that low levels of cholesterol in the CNS are associated with a decrease in synaptic function [17]. Some studies mention that serum cholesterol levels lower than 160 mg/dL (4.14 mmol/L) may affect human behavior through the serotonin system [18], which is related to a number of suicide attempts and violent deaths, in contrast with individuals who show higher serum cholesterol levels [19]. Moreover, some studies have shown that individuals with low serum cholesterol levels have a lower risk of mortality from heart disease. However, these patients presented mortality from non-cardiac diseases including suicide [20,21]. It is important to note that serum cholesterol levels are related to other mental health problems (depression, anxiety, schizophrenia, and bipolar disorder); therefore, it has been suggested that cholesterol levels can be considerate as a biomarker of these diseases [17,18,22]. There is a report that shows that low levels of total serum cholesterol (TC), LDL cholesterol, and C-reactive protein were significantly associated with suicide retries [23]. Aguglia et al. (2019) evaluated metabolic variables in different groups of patients and it was observed that the group of patients with a high-lethality suicide attempt had significantly lower levels of total cholesterol and HDL-c, and high levels C-reactive protein with respect to the control group. The authors propose that dyslipidemias contribute to changes in the membranes of serotonergic neurons and may contribute to suicidal behavior [23].

Several authors have reported that low serum cholesterol levels are related to violent behavior in suicide attempts [11,19,24]. This finding is in agreement with a study conducted in a group of patients who used violent methods when attempting suicide and had lower cholesterol levels, compared with individuals who attempted suicide by non-violent methods [24]. In the gray matter of the frontal cortex of people who completed suicide violently, lower levels of cholesterol were found in comparison with those subjects who used non-violent forms of suicide [17]. Contrarily, Atmaca et al. suggested that cholesterol and leptin levels have a connection with violent methods in suicide attempts [25]. Data in the existing literature propose that violent methods could be related to lipid metabolism [19,26], a possible mechanism has been proposed that suggests that a decrease in LDL levels in the CNS causes a decrease in the viscosity of the cell membrane; this leads to a decrease in 5-HT1A (serotonin) receptors, which in turn leads to impulsivity and violent suicidal behavior, secondary to low cholesterol levels [8,17,24,27]. Then, more studies with special attention to the relation between lower level of cholesterol and violence of suicide attempt should be performed.

In contrast, other studies suggest that the association between serum lipid concentrations and suicide attempt in patients with MDD is controversial and inconsistent [28]. Bartoli et al. (2017) [29] showed that subjects with a history of suicide attempt (174.0 ± 45.7 mg/dL) and without suicide attempts (193.9 ± 42.6 mg/dL) presented no significant difference in total cholesterol, LDL, or triglyceride levels [28,30]. However, other researchers have suggested that patients who attempted suicide had higher concentrations of CT, LDL-C, and TG than those who did not attempt suicide [31], besides, MDD subjects who attempted suicide had significantly higher HDL-C levels [32]. On the other hand, Piyal et al. (2020) observed low levels of TC, LDL-C, and HDL-C in patients with schizophrenia who presented violent behavior compared to subjects with major depressive disorder, bipolar affective disorder with suicide attempt. Conversely, patients with schizophrenia spectrum disorders who attempted suicide show low levels of total cholesterol [33]. In this regard, Brett et al. (2021) [34] suggest that MDD and bipolar disorder BD are associated with metabolic dysfunction of lipid, fatty acid, glucose regulation [34], and impaired glucose metabolism (obesity, metabolic syndrome) has been associated with the suicide attempt [14,31]. This might be partly due to the relationship between cholesterol metabolism and the serotonergic pathway [14,31,33,34].

All these findings together show that suicidal behavior is related with brain cholesterol that is responsible for the regulation of several processes: membrane-bound proteins, ion channels, synaptic transmission, synapse formation, dendritic formation, and the formation of axons; whose disturbances have been associated with mood disorders. Moreover, cholesterol participates in cell signaling. In this regard, the Sony Hedgehog (SHH) genes have an important role in the formation of many nervous structures [35]. The importance can be observed in the Smith-Lemli Opitz syndrome, an innate error of cholesterol metabolism, characterized by severe alterations in the central nervous system. This alteration is caused by a blockage of the dehydrocholesterol 7-alpha-reductase enzyme, which produces a significant decrease in cholesterol. Some research suggests an increase in suicidal behavior in patients affected by this genetic disease [36].

In this regard, diverse studies have suggested the association between dyslipidemia and suicidal behavior [37]. Additionally, other lipids such as polyunsaturated fatty acids (PUFAs) are associated with a beneficial effect on suicidal behavior [38]. Moreover, lipid composition is affected by cell function (lipid peroxidation, oxidative stress) and diet. Meanwhile, lipid peroxidation leads to oxidative deterioration of lipids, altering the permeability and fluidity of the membrane lipid degradation [39]. Some authors reported that levels of malondialdehyde (MDA) are increased in patients with depression [40,41].

## 3. Proposed Models Connecting Cholesterol Reduction with Suicide

### 3.1. Neuroinflammation

One theory of how cholesterol influences suicidal behavior is that of neuroinflammation (Figure 1). Low cholesterol levels could cause functional consequences in the lipid raft, these structures are microstructures enriched with cholesterol, sphingolipids, saturated fatty acids. and gangliosides. This disproportionate lipid rafts reacts with other cytokines promoting a process of inflammation. It is considered that n-3 and Toll-like receptors (TLR) are pattern recognition receptors that induce an inflammatory response by the nuclear factor Kappa by means of cytosine production. Moreover, TLRs are modulated by lipid rafts, and N-3 has an anti-inflammatory effect. However, when cholesterol levels decrease, the ratio PUFA N-6: N-3 becomes unbalanced, N-6 increases and leads to the production of pro-inflammatory cytokines, particularly IL-6, increasing the inflammatory process [42] (Figure 1). In accordance with the above, Aguglia et al. (2021) reported the possible relationship of platelets in the inflammatory response in individuals with a suicide attempt [43]. In addition, the participation of cholesterol, LDL, VLDL, and circulating chylomicrons in the vessel walls can be oxidized, triggering an immunological cascade, reactive oxygen species (ROS), and immune cells (macrophages, natural killer cells, mast cells or dendritic cells) [34]. Many of these pro-inflammatory molecules cross the blood–brain barrier causing psychiatric symptoms and mood disorders [34].

### 3.2. Serotonin

The second theory is in relation to the serotoninergic system. A decrease of cholesterol in the lipid rafts produces a decrease in the viscosity of the neuronal membranes. Low viscosity leads to a failure in the synaptic transmission, causing a decrease of serotonin intake via serotonin receptors (5-HT1A) [44]. It is known that among the serotonin functions are mood regulation. Furthermore, low levels of serotonin in the spinal fluid brain (CSF) have been observed in the postmortem of individuals who died by suicide [17]. Therefore, it might be that serum cholesterol levels reflect the serotoninergic activity [11]. Furthermore, fatty acid deficiency promotes an impaired serotonin and dopamine neurotransmission in the frontal cortex [34].

Additionally, for serotonin hypofunction, two other mechanisms have been proposed to explain increased impulsivity in individuals with suicide behavior: steroid modulation of serotonin communication and serotonin–dopamine interaction, but is possible that direct or indirect changes in lipid rafts may affect these mechanisms [11,45]. In this sense, various studies suggest that lipid rafts regulate dopaminergic and serotonergic neurotransmission [46], signal transduction [47], and interact with membrane-bound enzymes and ion channels. These findings suggest that the decrease in cholesterol and fatty acids ω3 could affect the formation of lipid rafts in the CNS and reduce in serotonin transmission and neuroinflammation [43,48]. Similarly, it has been hypothesized that the interaction between low serotonin and high testosterone (cholesterol-derivative) in the brain, mediate aggressive behavior and that the corticosteroids participate in suicide via a downregulation of 5-HT1A receptors (Figure 1) [49,50].

## 4. Genetics of Cholesterol Regulation and Suicide

On the other hand, due to the relevant findings involving serum cholesterol levels in suicidal behavior, some genes related to cholesterol synthesis and their possible involvement with suicide have been studied (HMGCRM, DHCR7, LPL, LDLR, and ApoE genes) [51]. Recently, ApoE levels in CSF and plasma have been associated with death by suicide [52,53]. Similarly, ApoE (ApoE4) isoforms have been related to neurodegenerative diseases, decreased recycling, and cholesterol redistribution in the central nervous system [54]. In that sense, a study in a Mexican population found that ApoE4 carriers have 4.92 times higher suicide odds (*p* = 0.0006) compared to individuals not carrying this allele [55].

Cholesterol and phospholipids in the brain are important in synapse maintenance and function, a high expression of genes involved in lipid trafficking suggests an association between cholesteryl ester hydrolase and violent suicides in the prefrontal cortex [10]. A greater expression of another gene, ACP1, was identified in cerebral tissue of individuals who died by suicide and co-expression analysis suggested that it is important to the regulation of brain mechanisms linked to suicide, including cholesterol synthesis, β-catenin-mediated signaling pathway, serotonin, GABA, and the stress response via ARHGAP35 [56,57].

Polymorphisms of this gene have been associated with suicide behavior in GWAS studies, particularly rs300774 and rs7419262 SNPs. The rs300774 variant might act as trans for genes related to cholesterol synthesis [57], whereas, rs300774 has been associated with suicide attempt [58]. Nevertheless, evidence obtained so far is not enough to understand the role of these polymorphisms in individuals with suicidal behavior. Therefore, more studies assessing the relation between genes in the cholesterol pathway in patients with psychiatric disorders are needed [59,60].

## 5. Pharmacological Treatments That Alter Lipid Metabolism and the Possible Participation in Suicidal Behavior

Some drugs used in the treatment of mental illnesses can modify metabolic parameters. One of the known drugs with this effect is olanzapine, which may modify the levels of cholesterol, triglycerides, and glucose [61]. Liu et al. (2019) showed that olanzapine increased serum cholesterol and statin administration achieved a reduction in serum cholesterol in rats. The authors suggest that it is possibly through regulation of protein SREBP1 [62]. It has been observed in hypercholesterolemic rats that atorvastatin can increase the concentration of cholesterol in the spinal cord and decreases serum levels [63]. Although, the biochemical mechanisms for understanding cholesterol transport in the central nervous system have not been elucidated.

Other reports have linked statin use to depressive symptoms; mainly lipophilic statins, possibly because these statins penetrate many tissues more easily; people who used statins had a prevalence of up to 2.5 times higher prevalence of suicidal behavior than the group that did not receive statin treatment; they also evaluated the effect of a diet with high fat intake, and in this variable no relationship was found [64]. For his part, Harro (2018) showed that rodents treated with simvastine showed a relationship between aggressive behavior, anxiety, and cholesterol reduction [65]. Together, these findings suggest a possible relationship between depressive disorders, suicide, and low cholesterol levels, because these drugs inhibit cholesterol synthesis in the liver and decreased serum levels, cholesterol, and omega-3 PUFAs and statins have been shown to be anti-inflammatory. However, further studies in animal models are required to support and reinforce this hypothesis.

## 6. Conclusions

In conclusion, the literature suggests an association between individuals with lower cholesterol levels and suicidal behavior, this may be caused by changes that induce failures in synaptic transmission or because low cholesterol levels seem to induce an inflammatory response in the nervous system. Additionally, lower serum cholesterol levels have been associated with violent methods used in suicide attempts. The presence of the ApoE4 allele, which has been associated with less cholesterol redistribution in the central nervous system, should also be considered, due to the mechanisms already described. Therefore, more research is needed to understand the mechanism that associates serum cholesterol levels and suicidal behavior. In addition, research on how serum cholesterol affects CNS cholesterol metabolism should be conducted to fully establish the role of cholesterol in suicidal behavior.

## Figures and Tables

**Figure 1 biomedicines-09-01559-f001:**
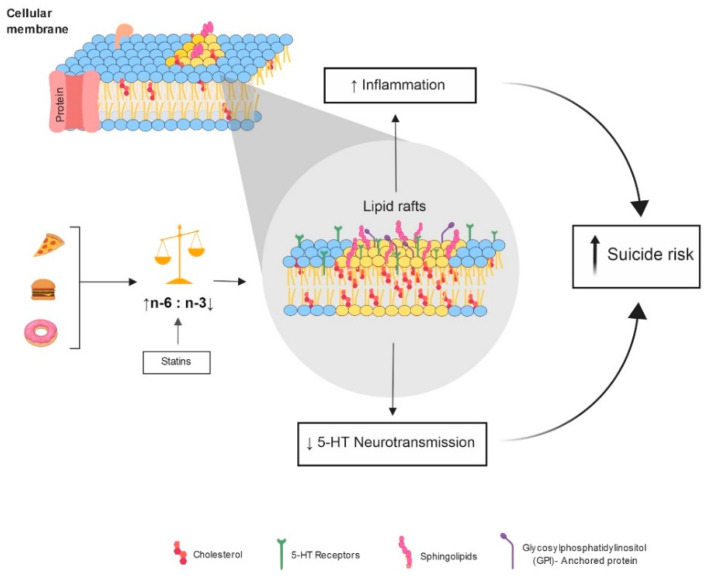
Theories of two biologic mechanisms that associate cholesterol and suicidal behavior. Abbreviations: 5-HT: 5-hydroxytryptamine receptors; GPI: glycosylphosphatidylinositol—anchored protein; n-3: omega 3; n-6: omega 6; PUFA: polyunsaturated fatty acids; ↑, increased; ↓, decreased.

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
