# Peer review of "Possible Association of Cholesterol as a Biomarker in Suicide Behavior"

_biomedicines, 2021, doi:10.3390/biomedicines9111559_

Round 1
Reviewer 1 Report
- The abstract does not reflect the topic and conclusions presented.
- The introduction - This is not an article about suicides; too long and unnecessary introduction.
- Association between cholesterol and suicidality
This chapter is a mixture of different pieces of information. There is no meaningful division (cholesterol levels, nextly biochemistry, etc. - no logical division)
- Too cursory description of a lot of information, e.g.:
“Several researches have related low serum cholesterol levels with the violence of the suicide attempt, but not with death by suicide [21, 27, 38].” (This is just an example)
- The main factor prompting me to reject article
It should be a systematic review or meta-analysis. This would increase the value and clarity of the manuscript. An example of such an article: Lester, D. Serum cholesterol levels and suicide: a meta-analysis. Suicide Life Threat Behav 2002, 32, 333-46
The authors do not cite important reports on the topic! Examples (Authors should prepare this type of article – systematic review and meta-analysis):
https://pubmed.ncbi.nlm.nih.gov/33301469/
https://pubmed.ncbi.nlm.nih.gov/33834474/
https://pubmed.ncbi.nlm.nih.gov/34240626/
https://pubmed.ncbi.nlm.nih.gov/33477435/ - The authors did not even cite such an important article.
Other important example of article: https://pubmed.ncbi.nlm.nih.gov/34440563/
- Pharmacological treatments with lipid-lowering drugs and suicidal behavior
The authors wrote too generally, sketchily. Examples:
“In contrast, there is evidence indicating that there is no clear association”
“Other studies have linked the use of statins with possible depressive symptoms; mainly lipophilic statins”
The authors do not provide specific data, do not describe the quoted citations in detail.
I am compelled to recommend the rejection of the manuscript. There are many important articles on the same topic (including meta-analyzes) that the authors have not cited (examples are given). The submitted article is nothing innovative. When submitting an article to another journal, authors should perform a meta-analysis. Tables should be created with data such as: cholesterol level, results of analyzes, risk of suicide, etc. The submitted article is not suitable for Biomedicines, it is too low quality. Biomedicines publishes high-quality systematic reviews and meta-analyzes, not just a review of few articles.
Author Response
REVIEWER 1
Comment 1: The abstract does not reflect the topic and conclusions presented.
Replay: We appreciate this observation and make changes to the section abstract.
Change in the manuscript:
- Abstract: Suicides and suicidal behavior are major causes of mortality and morbidity in public health and are a global problem. Various authors have proposed that changes in lipid metabolism (total cholesterol decrease) as a possible biological marker for suicidal behavior. The objective of this study was to review the studies that have demonstrated a relationship between se-rum cholesterol levels and suicidal behavior and to describe the possible pathophysiological mechanisms that associate changes in cholesterol concentration and suicidal behavior. Relevant literature related to serum cholesterol levels and suicidal behavior was identified through various database searches. The data from the existing literature present the findings that relate low cholesterol levels and possible pathophysiological mechanisms (neuroinflammation, serotonergic neurotransmission), genes related to cholesterol synthesis, pharmacological treatments that alter lipid metabolism and the possible participation in suicidal behavior. Nevertheless, future research is required to describe how serum cholesterol affects cholesterol metabolism in the CNS to establish and understand the role of cholesterol in suicidal behavior… Abstract Page 1, Line 25
Comment 2: The introduction - This is not an article about suicides; too long and unnecessary introduction.
Replay: Thank you very much. We agree with your comment. We make changes to the introduction section according to your valuable suggestion.
Change in the manuscript:
- There are many variables that are related to suicidal behavior, some articles even propose an important role of climate change and environmental factors [3]… Introduction Page 2, Line 48-50
- Cholesterol is a lipid that realizes many important functions in the cell, one of them is cell signaling, since cholesterol is essential for lipid rafts, structures necessary for multiple cellular functions [4]. Alterations in cholesterol metabolism mainly affect cardiovascular diseases, but also neurodegenerative diseases and even cancer, due to the role that cholesterol plays in metabolism [5]. The possible role of cholesterol in suicidal behavior is currently being investigated, however, the results are controversial, but many of these studies did not consider confounding variables that can affect cholesterol metabolism… Introduction Page 2, Line 51-57
Comment 3: Association between cholesterol and suicidality.
This chapter is a mixture of different pieces of information. There is no meaningful division (cholesterol levels, nextly biochemistry, etc. - no logical division).
Replay: We try to improve our manuscript according to the suggestions of reviewers 1, 2 and 3. Hence, we modified some particular sections of our work.
Change in the manuscript:
- Several studies have investigated a possible link between low serum cholesterol and suicide. Zhang and et al. (2020) suggest that MDD patients who attempted suicide had significantly lower serum Total cholesterol and LDL-C levels than those who did not attempt suicide. In this sense, Wu et al. (2016) [13] by means of a meta-analysis study in patients with depressive disorders (depression, schizoaffective depression, major depressive episodes and MDD), they report that patients with suicide attempt have lower serum levels of TC, TG HDL-C, LDL-C than non-suicidal patients. Besides, the lower serum concentrations of TC present a relationship with the suicide attempt in middle-aged and older patients with respect to subjects under 40 years of age [14]. These findings together support the hypothesis that lower serum concentrations of CT and LDL-C are associated with suicide attempt in patients with MDD. In a study carried out in 52 patients hospitalized after a suicide attempt, several parameters were analyzed, a statistically significant association was found with the decrease in total cholesterol and HDL-C, a relationship was also observed between the decrease in total serum cholesterol and hostile personality traits [15]. Lester et al. 2002 [16] found that people with lower levels of serum cholesterol had a statistically significant rate of completion of suicide, compared to people who had serum cholesterol in normal ranges, and people who had attempted suicide in the past also had lower cholesterol levels, especially when they used violent suicide methods… Page 2,3, Line 88-105
- Besides, some studies have shown that individuals with low serum cholesterol levels have a lower risk of mortality from heart disease. However, these patients presented mortality from non-cardiac diseases including suicide [20, 21]…. Page 2,3, 112-115
- There is a report that shows that low levels of total serum cholesterol (TC), LDL cholesterol and C-reactive protein were significantly associated with suicide retries [23]. Aguglia et al., (2019) evaluated metabolic variables in different groups of patients and it was observed that the group of patients with a high-lethality suicide attempt had significantly lower levels of total cholesterol and HDL-c, and high levels C-reactive protein with respect to the control group. The authors propose that dyslipidemias contribute to changes in the membranes of serotonergic neurons, and may contribute to suicidal behavior [23]… Page 3, 118-125
- Several authors have reported that low serum cholesterol levels are related to violent behavior in suicide attempts [11, 19, 24]. This finding is in agreement with a study con-ducted in a group of patients who used violent methods when attempting suicide and had lower cholesterol levels, compared with individuals who attempted suicide by non-violent methods [24]. In the gray matter of the frontal cortex of people who completed suicide violently, lower levels of cholesterol were found in comparison with those subjects who used non-violent forms of suicide [17]. Contrarily, Atmaca et al., suggested that cholesterol and leptin levels have a connection with violent methods in suicide attempts [25]. Data in the existing literature propose that violent could be related to lipid metabolism [19, 26], a possible mechanism has been proposed that suggests that a decrease in LDL levels in the CNS causes a decrease in the viscosity of the cell membrane; and this leads to a decrease in 5-HT1A (serotonin) receptors, which in turn leads to impulsivity and violent suicidal behavior, secondary to low cholesterol levels [8, 17, 24, 27]. Then, more studies with special attention to the relation between lower level of cholesterol and violence of suicide at-tempt should be performed… Page 3, 126-140
- In contrast, other studies suggest that the association between serum lipid concentrations and suicide attempt in patients with MDD is controversial and inconsistent [28]. Bartoli et al. (2017) [29] showed that subjects with a history of suicide attempt (174.0 ± 45.7 mg / dL) and without suicide attempts (193.9 ± 42.6 mg / dl) presented no significant difference in total cholesterol, LDL, or triglyceride levels [28, 30]. However, other researchers have suggested that patients who attempted suicide had higher concentrations of CT, LDL-C, and TG than those who did not attempt suicide [31], besides, MDD subjects who attempted suicide had significantly higher HDL-C levels [32]. On the other hand, Piyal et al., (2020) observed low levels of TC, LDL-C and HDL-C in patients with schizophrenia who presented violent behavior than subjects with major depressive disorder, bipolar affective disorder with suicide attempt. Conversely, patients with schizophrenia spectrum disorders who attempted suicide show low levels of total cholesterol [33]. With regard, Brett et al., (2021) [34] suggest that MDD and Bipolar Disorder BD are associated with metabolic dysfunction of lipid, fatty acid, glucose regulation [34], and impaired glucose metabolism (obesity, metabolic syndrome) has been associated with the suicide attempt [14, 31]. This might be partly due to the relationship between cholesterol metabolism and the serotonergic pathway [14, 31, 33, 34]… Page 3,4, 141-157
- All these findings together show that suicidal behavior is related with brain choles-terol is responsible for the regulation of several processes: membrane-bound proteins, ion channels, synaptic transmission, synapse formation, dendritic formation and the for-mation of axons; whose disturbances have been associated with mood disorders… Page 4, 158-161
Comment 4: Too cursory description of a lot of information, e.g.: “Several researches have related low serum cholesterol levels with the violence of the suicide attempt, but not with death by suicide [21, 27, 38]” (This is just an example).
Replay: We acknowledge and apologize for the old version. However, thanks to your comment, we have worked on and improved this section of the manuscript.
Change in the manuscript:
- Several authors have reported that low serum cholesterol levels are related to violent behavior in suicide attempts [11, 19, 24]. This finding is in agreement with a study con-ducted in a group of patients who used violent methods when attempting suicide and had lower cholesterol levels, compared with individuals who attempted suicide by non-violent methods [24]. In the gray matter of the frontal cortex of people who completed suicide vio-lently, lower levels of cholesterol were found in comparison with those subjects who used non-violent forms of suicide [17]. Contrarily, Atmaca et al., suggested that cholesterol and leptin levels have a connection with violent methods in suicide attempts [25]. Data in the existing literature propose that violent could be related to lipid metabolism [19, 26], a pos-sible mechanism has been proposed that suggests that a decrease in LDL levels in the CNS causes a decrease in the viscosity of the cell membrane; and this leads to a decrease in 5-HT1A (serotonin) receptors, which in turn leads to impulsivity and violent suicidal behavior, secondary to low cholesterol levels [8, 17, 24, 27]. Then, more studies with spe-cial attention to the relation between lower level of cholesterol and violence of suicide at-tempt should be performed… Page 3, 126-140
Comment 5: The main factor prompting me to reject article.
It should be a systematic review or meta-analysis. This would increase the value and clarity of the manuscript. An example of such an article: Lester, D. Serum cholesterol levels and suicide: a meta-analysis. Suicide Life Threat Behav 2002, 32, 333-46
The authors do not cite important reports on the topic! Examples (Authors should prepare this type of article – systematic review and meta-analysis):
https://pubmed.ncbi.nlm.nih.gov/33301469/
https://pubmed.ncbi.nlm.nih.gov/33834474/
https://pubmed.ncbi.nlm.nih.gov/34240626/
https://pubmed.ncbi.nlm.nih.gov/33477435/ - The authors did not even cite such an important article.
Replay: We agree with your observation. For a future study we are making a record in PROSPERO for a systematic review and meta-analysis. However, in the present study only a review was had as a priori objective. On the other hand, we appreciate the suggested manuscripts. We have added these references that enrich our work. Therefore, we made changes to the manuscript that are listed below.
Change in the manuscript:
- Several studies have investigated a possible link between low serum cholesterol and suicide. Zhang and et al. (2020) suggest that MDD patients who attempted suicide had significantly lower serum Total cholesterol and LDL-C levels than those who did not attempt suicide. In this sense, Wu et al. (2016) [13] by means of a meta-analysis study in patients with depressive disorders (depression, schizoaffective depression, major depressive episodes and MDD), they report that patients with suicide attempt have lower serum levels of TC, TG HDL-C, LDL-C than non-suicidal patients. Besides, the lower serum concentrations of TC present a relationship with the suicide attempt in middle-aged and older patients with respect to subjects under 40 years of age [14]. These findings together support the hypothesis that lower serum concentrations of CT and LDL-C are associated with suicide attempt in patients with MDD… Page 2, Lines 88-98
- In contrast, other studies suggest that the association between serum lipid concentra-tions and suicide attempt in patients with MDD is controversial and inconsistent [28]. Bartoli et al. (2017) [29] showed that subjects with a history of suicide attempt (174.0 ± 45.7 mg / dL) and without suicide attempts (193.9 ± 42.6 mg / dl) presented no significant dif-ference in total cholesterol, LDL, or triglyceride levels [28, 30]. However, other researchers have suggested that patients who attempted suicide had higher concentrations of CT, LDL-C, and TG than those who did not attempt suicide [31], besides, MDD subjects who attempted suicide had significantly higher HDL-C levels [32]. On the other hand, Piyal et al., (2020) observed low levels of TC, LDL-C and HDL-C in patients with schizophrenia who presented violent behavior than subjects with major depressive disorder, bipolar af-fective disorder with suicide attempt. Conversely, patients with schizophrenia spectrum disorders who attempted suicide show low levels of total cholesterol [33]. With regard, Brett et al., (2021) [34] suggest that MDD and Bipolar Disorder BD are associated with metabolic dysfunction of lipid, fatty acid, glucose regulation [34], and impaired glucose metabolism (obesity, metabolic syndrome) has been associated with the suicide attempt [14, 31]. This might be partly due to the relationship between cholesterol metabolism and the serotonergic pathway [14, 31, 33, 34]… Page 3,4, Lines 141-157
- All these findings together show that suicidal behavior is related with brain choles-terol is responsible for the regulation of several processes: membrane-bound proteins, ion channels, synaptic transmission, synapse formation, dendritic formation and the for-mation of axons; whose disturbances have been associated with mood disorders… Page 4, Lines 158-161
Comment 5.1: The authors did not even cite such an important article. Other important example of article: https://pubmed.ncbi.nlm.nih.gov/34440563/
Replay: Thank you very much. We include this interesting reference in the manuscript.
Change in the manuscript:
- Besides, the participation of cholesterol, LDL, VLDL and circulating chylomicrons in the vessel walls can be oxidized, triggering an immunological cascade, reactive oxygen species (ROS) and immune cells (macrophages, natural killer cells, mast cells or dendritic cells [34]. Many of these pro-inflammatory molecules cross the blood-brain barrier causing psychiatric symptoms and mood disorders [34]… Page 4, Lines 193-197
- Furthermore, fatty acid deficiency promotes an impaired serotonin and dopamine neuro-transmission in the frontal cortex [34]… Page 5, Lines 207-208
Comment 6: Pharmacological treatments with lipid-lowering drugs and suicidal behavior.
Replay: Thank you for your valuable comments that have helped us improve the quality of our work. We are indebted to you and the other reviewers. The comments corresponding to this section are detailed in the next questions. On the other hand, we have modified the name of this section.
Change in the manuscript:
- Pharmacological treatments that alter lipid metabolism and the possible participation in suicidal behavior… Page 6, Lines 250-251
Comment 6.1: The authors wrote too generally, sketchily. Examples: “In contrast, there is evidence indicating that there is no clear association” “Other studies have linked the use of statins with possible depressive symptoms; mainly lipophilic statins”
Replay: We appreciate the observation and we make changes all manuscript. Also, due to the comment of the reviewers, we have to change or eliminate some information in the manuscript.
Change in the manuscript (an example):
- Other reports have linked statin use to depressive symptoms; mainly lipophilic statins, possibly because these statins penetrate many tissues more easily; people who used statins had a prevalence of up to 2.5 times higher prevalence of suicidal behavior than the group that did not receive statin treatment; they also evaluated the effect of a diet with high fat intake, and in this variable no relationship was found [64]…. Page 6, Lines 261-265
- Comment deleted: In contrast, there is evidence indicating that there is no clear association… Page 6, Line 267
Comment 6.2: The authors do not provide specific data, do not describe the quoted citations in detail.
Replay: We appreciate your comments and take in consideration in order to improve the quality of the manuscript. We discussed in a detail more specific information.
Change in the manuscript:
- Several studies have investigated a possible link between low serum cholesterol and suicide. Zhang and et al. (2020) suggest that MDD patients who attempted suicide had significantly lower serum Total cholesterol and LDL-C levels than those who did not at-tempt suicide. In this sense, Wu et al. (2016) [13] by means of a meta-analysis study in pa-tients with depressive disorders (depression, schizoaffective depression, major depressive episodes and MDD), they report that patients with suicide attempt have lower serum lev-els of TC, TG HDL-C, LDL-C than non-suicidal patients. Besides, the lower serum concen-trations of TC present a relationship with the suicide attempt in middle-aged and older patients with respect to subjects under 40 years of age [14]. These findings together sup-port the hypothesis that lower serum concentrations of CT and LDL-C are associated with suicide attempt in patients with MDD. In a study carried out in 52 patients hospitalized after a suicide attempt, several parameters were analyzed, a statistically significant association was found with the decrease in total cholesterol and HDL-C, a relationship was also observed between the decrease in total serum cholesterol and hostile personality traits [15]. Lester et al. 2002 [16] found that people with lower levels of serum cholesterol had a statistically significant rate of completion of suicide, compared to people who had serum cholesterol in normal ranges, and people who had attempted suicide in the past also had lower cholesterol levels, especially when they used violent suicide methods… Page 2,3, Lines 88-105
- Aguglia et al., (2019) evaluated metabolic variables in different groups of patients and it was observed that the group of patients with a high-lethality suicide attempt had signifi-cantly lower levels of total cholesterol and HDL-c, and high levels C-reactive protein with respect to the control group. The authors propose that dyslipidemias contribute to changes in the membranes of serotonergic neurons, and may contribute to suicidal behavior [23]… Page 2,3, Lines 120-125
- Several authors have reported that low serum cholesterol levels are related to violent behavior in suicide attempts [11, 19, 24]. This finding is in agreement with a study con-ducted in a group of patients who used violent methods when attempting suicide and had lower cholesterol levels, compared with individuals who attempted suicide by non-violent methods [24]. In the gray matter of the frontal cortex of people who completed suicide vio-lently, lower levels of cholesterol were found in comparison with those subjects who used non-violent forms of suicide [17]. Contrarily, Atmaca et al., suggested that cholesterol and leptin levels have a connection with violent methods in suicide attempts [25]. Data in the existing literature propose that violent could be related to lipid metabolism [19, 26], a pos-sible mechanism has been proposed that suggests that a decrease in LDL levels in the CNS causes a decrease in the viscosity of the cell membrane; and this leads to a decrease in 5-HT1A (serotonin) receptors, which in turn leads to impulsivity and violent suicidal behavior, secondary to low cholesterol levels [8, 17, 24, 27]. Then, more studies with spe-cial attention to the relation between lower level of cholesterol and violence of suicide at-tempt should be performed… Page 3, Lines 126-140
- In contrast, other studies suggest that the association between serum lipid concentrations and suicide attempt in patients with MDD is controversial and inconsistent [28]. Bartoli et al. (2017) [29] showed that subjects with a history of suicide attempt (174.0 ± 45.7 mg / dL) and without suicide attempts (193.9 ± 42.6 mg / dl) presented no significant difference in total cholesterol, LDL, or triglyceride levels [28, 30]. However, other researchers have suggested that patients who attempted suicide had higher concentrations of CT, LDL-C, and TG than those who did not attempt suicide [31], besides, MDD subjects who attempted suicide had significantly higher HDL-C levels [32]. On the other hand, Piyal et al., (2020) observed low levels of TC, LDL-C and HDL-C in patients with schizophrenia who presented violent behavior than subjects with major depressive disorder, bipolar affective disorder with suicide attempt. Conversely, patients with schizophrenia spectrum disorders who attempted suicide show low levels of total cholesterol [33]. With regard, Brett et al., (2021) [34] suggest that MDD and Bipolar Disorder BD are associated with metabolic dysfunction of lipid, fatty acid, glucose regulation [34], and impaired glucose metabolism (obesity, metabolic syndrome) has been associated with the suicide attempt [14, 31]. This might be partly due to the relationship between cholesterol metabolism and the serotonergic pathway [14, 31, 33, 34]… Page 3, 4, Lines 141-157
- Other reports have linked statin use to depressive symptoms; mainly lipophilic statins, possibly because these statins penetrate many tissues more easily; people who used statins had a prevalence of up to 2.5 times higher prevalence of suicidal behavior than the group that did not receive statin treatment; they also evaluated the effect of a diet with high fat intake, and in this variable no relationship was found [64]… Page 6, Lines 261-265

Reviewer 2 Report
Introduction: Lines 68-70: authors refer the risk for death by suicide in patients with type 2 diabetes. Why is that so? How does this phatological condition affects the body / psyche that increases this rate?
Figure 1: legend are decribing events on A) B) and C) but this references are not presented in the figure.
Line 93 acronym ER is presented withou any reference.
The low colesterol level and its physiological properties are explained but which typoe of cholesterol is investigated? Is it LDL, HDL?
Overall in text authors report associations found in other studies (e.g. Lines 215-216). Associations don't lead to causality. It would be more informative to the reader to know how (big?) this associations are and if there were other findings regarding the relationships about cholesterol and suicide (besides association).
Also sentences with "hogher propability (line 219" or higher risk, should be described precisely. How higher is this risk? Is this difference statsitically relevant? It would be recommendable to only report findings were such higher risks are statistically significant and then present those statistics.
Conclusion: "In conclusion, the literature suggests an association between individuals with lower levels of cholesterol and suicide behavior". Again, associations are no causality. The reader is looking for learning about causalities. What is it that actually influences death by suicide? What are the proven facts - besides associations - that were reported?
Authors should rewrite the manuscript and avoid to report only associations, and if so please insert which associations they are referring to and if those are statistically relevant for the audience.
Author Response
REVIEWER 2
Comment 1: Introduction: Lines 68-70: authors refer the risk for death by suicide in patients with type 2 diabetes. Why is that so? How does this phatological condition affects the body / psyche that increases this rate?
Replay: We apologize. We recognized that the data is confused; hence we perform some modified the introduction section.
Change in the manuscript:
- Term suicide encompasses ideation, intent, and actions. Suicidal ideation is defined as thinking about or considering suicide; ranging from passive death wishes to active thoughts of committing suicide. In this sense, Suicide is listed by the World Health Organization as one of the leading causes of deaths worldwide. About 800 000 people take their own lives every year, which generates a global mortality rate of 16 per 100 000 inhabitants. Lifetime prevalence rates are approximately 9.2% for suicidal ideation and 2.7% for suicide attempts [1]. It is estimated that at least 20 people try to commit suicide per each individual who dies by suicide [2]. There are many variables that are related to suicidal behavior, some articles even propose an important role of climate change and environ-mental factors [3]… Introduction Page 1,2, Lines 30-50
- Cholesterol is a lipid that realizes many important functions in the cell, one of them is cell signaling, since cholesterol is essential for lipid rafts, structures necessary for multiple cellular functions [4]. Alterations in cholesterol metabolism mainly affect cardiovascular diseases, but also neurodegenerative diseases and even cancer, due to the role that cholesterol plays in metabolism [5]. The possible role of cholesterol in suicidal behavior is currently being investigated, however, the results are controversial, but many of these studies did not consider confounding variables that can affect cholesterol metabolism… Introduction Page 2, Lines 51-57
Comment 2: Figure 1: legend are decribing events on A) B) and C) but this references are not presented in the figure.
Replay: Thank you very much. According to the comments obtained in this review process, we have to deleted the Figure 1.
Change in the manuscript:
- Figure 1 removed.
Comment 3: Line 93 acronym ER is presented without any reference.
Replay: Done. We appreciate your comment. We have explicitly added the information.
Change in the manuscript:
- Endoplasmatic reticulum (ER)… Introduction Page 2, Lines 71-72
Comment 4: The low cholesterol level and its physiological properties are explained but which type of cholesterol is investigated? Is it LDL, HDL?
Replay: Done. We appreciate your question because it helps us to have clarity for the readers of the manuscript.
Change in the manuscript:
- Total cholesterol is investigated, and some authors also propose HDL-c, it may be related to suicidal behavior.
Comment 5: Overall in text authors report associations found in other studies (e.g. Lines 215-216). Associations don't lead to causality. It would be more informative to the reader to know how (big?) this association are and if there were other findings regarding the relationships about cholesterol and suicide (besides association).
Replay: We follow the suggestion of been more informative to reader. We modify it throughout the manuscript. As examples:
Change in the manuscript:
- There is a report that shows that low levels of total serum cholesterol (TC), LDL cholesterol and C-reactive protein were significantly associated with suicide retries [23]. Aguglia et al., (2019) evaluated metabolic variables in different groups of patients and it was observed that the group of patients with a high-lethality suicide attempt had significantly lower levels of total cholesterol and HDL-c, and high levels C-reactive protein with respect to the control group. The authors propose that dyslipidemias contribute to changes in the membranes of serotonergic neurons, and may contribute to suicidal behavior [23]… Page 3, Lines 118-125
- Lester et al. 2002 [16] found that people with lower levels of serum cholesterol had a sta-tistically significant rate of completion of suicide, compared to people who had serum cholesterol in normal ranges, and people who had attempted suicide in the past also had lower cholesterol levels, especially when they used violent suicide methods… Page 3, Lines 102-105
Comment 6: Also sentences with "higher probability (line 219" or higher risk, should be described precisely. How higher is this risk? Is this difference statistically relevant? It would be recommendable to only report findings were such higher risks are statistically significant and then present those statistics.
Replay: Thanks. We have specified the information described.
Change in the manuscript:
- In that sense, a study in a Mexican population found that ApoE4 carriers have 4.92 times higher suicide odds (p=0.0006) compared to individuals not carrying this allele [55]… Page 5, Lines 233-234
Comment 7: Conclusion: "In conclusion, the literature suggests an association between individuals with lower levels of cholesterol and suicide behavior". Again, associations are no causality. The reader is looking for learning about causalities. What is it that actually influences death by suicide? What are the proven facts - besides associations - that were reported?
Replay: Thank you for your comment. The conclusion was modified, as suggested by the reviewer.
Change in the manuscript:
- In conclusion, the literature suggests an association between individuals with lower cholesterol levels and suicidal behavior, this may be caused by changes that induce failures in synaptic transmission or because low cholesterol levels seem to induce an inflammatory response in the nervous system. Additionally, lower serum cholesterol levels have been associated with violent methods used in suicide attempts. The presence of the ApoE4 allele, which has been associated with less cholesterol redistribution in the central nervous system, should also be considered, due to the mechanisms already described. Therefore, more research is needed to understand the mechanism that associates serum cholesterol levels and suicidal behavior. In addition, research on how serum cholesterol affects CNS cholesterol metabolism should be conducted to fully establish the role of cholesterol in suicidal behavior… Page 6, Lines 276-286
Comment 8: Authors should rewrite the manuscript and avoid to report only associations, and if so please insert which associations they are referring to and if those are statistically relevant for the audience.
Replay: Thank you for your comment. We have worked the entire manuscript to improve the quality.
- There is a report that shows that low levels of total serum cholesterol (TC), LDL cholesterol and C-reactive protein were significantly associated with suicide retries [23]. Aguglia et al., (2019) evaluated metabolic variables in different groups of patients and it was observed that the group of patients with a high-lethality suicide attempt had significantly lower levels of total cholesterol and HDL-c, and high levels C-reactive protein with respect to the control group. The authors propose that dyslipidemias contribute to changes in the membranes of serotonergic neurons, and may contribute to suicidal behavior [23]… Page 3, Lines 118-125
- Lester et al. 2002 [16] found that people with lower levels of serum cholesterol had a sta-tistically significant rate of completion of suicide, compared to people who had serum cholesterol in normal ranges, and people who had attempted suicide in the past also had lower cholesterol levels, especially when they used violent suicide methods… Page 3, Lines 102-105
- In that sense, a study in a Mexican population found that ApoE4 carriers have 4.92 times higher suicide odds (p=0.0006) compared to individuals not carrying this allele [55]… Page 5, Lines 233-234
- Other reports have linked statin use to depressive symptoms; mainly lipophilic statins, possibly because these statins penetrate many tissues more easily; people who used statins had a prevalence of up to 2.5 times higher prevalence of suicidal behavior than the group that did not receive statin treatment; they also evaluated the effect of a diet with high fat intake, and in this variable no relationship was found [64]… Page 6, Lines 261-265

Reviewer 3 Report
Many thanks to have the opportunity to revise this manuscript aiming to review the studies that have shown a relationship between cholesterol levels and suicidal behavior, in order to understand the possible pathophysiological mechanisms that associate changes in cholesterol concentration and suicidal behavior. This is very interesting and several evidence are presnet in recent literature that in my opinion should be added (see for example Frontiers in Psychiatry 2021; 12:653390 to insert in the introduction for the risk factor).
Regarding cholesterol and suicidal behaviours, I think that several recent references are missing and the clinical section should be addressed better (see for example cross-sectional study on Frontiers in Psychiatry 2019; 10:70, and a prospective study on J Affect Disord 2020; 271:328-335).
Related to serotonergic system another missing paper could be (World J Biol Psychiatry 2021; 22: 119-127).
Author Response
REVIEWER 3
Comment 1: Many thanks to have the opportunity to revise this manuscript aiming to review the studies that have shown a relationship between cholesterol levels and suicidal behavior, in order to understand the possible pathophysiological mechanisms that associate changes in cholesterol concentration and suicidal behavior. This is very interesting and several evidence are present in recent literature that in my opinion should be added (see for example Frontiers in Psychiatry 2021; 12:653390 to insert in the introduction for the risk factor).
Replay: We appreciate your comments. Thank you for the opportunity to work on and improve our manuscript. We are indebted to you and the other reviewers. We have added the suggested manuscript; it is very interesting.
Change in the manuscript:
- There are many variables that are related to suicidal behavior, some articles even propose an important role of climate change and environmental factors [3]… Page 2, Lines 48-50
Comment 2: Regarding cholesterol and suicidal behaviours, I think that several recent references are missing and the clinical section should be addressed better (see for example cross-sectional study on Frontiers in Psychiatry 2019; 10:70, and a prospective study on J Affect Disord 2020; 271:328-335).
Replay: Thank you for your comment. We have added this quote to our manuscript.
Change in the manuscript:
- There is a report that shows that low levels of total serum cholesterol (TC), LDL cholesterol and C-reactive protein were significantly associated with suicide retries [23]. Aguglia et al., (2019) evaluated metabolic variables in different groups of patients and it was ob-served that the group of patients with a high-lethality suicide attempt had significantly lower levels of total cholesterol and HDL-c, and high levels C-reactive protein with respect to the control group. The authors propose that dyslipidemias contribute to changes in the membranes of serotonergic neurons, and may contribute to suicidal behavior [23]… Page 3, Lines 118-125
Comment 3: Related to serotonergic system another missing paper could be (World J Biol Psychiatry 2021; 22: 119-127).
Replay: We appreciate your comment and it is included in the manuscript.
Change in the manuscript:
- In accordance with the above, Aguglia et al., (2021) has reported the possible relationship of platelets in the inflammatory response in individuals with a suicide attempt [43]… Page 4, Lines 191-193

Round 2
Reviewer 1 Report
The authors improved the manuscript significantly.